# Influences of Vitamin B_12_ Supplementation on Cognition and Homocysteine in Patients with Vitamin B_12_ Deficiency and Cognitive Impairment

**DOI:** 10.3390/nu14071494

**Published:** 2022-04-02

**Authors:** Asako Ueno, Tadanori Hamano, Soichi Enomoto, Norimichi Shirafuji, Miwako Nagata, Hirohiko Kimura, Masamichi Ikawa, Osamu Yamamura, Daiki Yamanaka, Tatsuhiko Ito, Yohei Kimura, Masaru Kuriyama, Yasunari Nakamoto

**Affiliations:** 1Second Department of Internal Medicine, Faculty of Medical Sciences, University of Fukui, 23-3 Matsuokashimoaizuki, Eiheiji-cho, Yoshida-gun, Fukui 910-1193, Japan; maedaa@u-fukui.ac.jp (A.U.); weltraum@u-fukui.ac.jp (S.E.); sira@u-fukui.ac.jp (N.S.); iqw@u-fukui.ac.jp (M.I.); kapi@u-fukui.ac.jp (O.Y.); dy0319@u-fukui.ac.jp (D.Y.); nakamoto-med2@med.u-fukui.ac.jp (Y.N.); 2Department of Neurology, Fukui-ken Saiseikai Hospital, 7-1 Funabashi, Wadanaka-cho, Fukui 918-8503, Japan; 3Department of Aging and Dementia (DAD), Faculty of Medical Sciences, University of Fukui, 23-3 Matsuokashimoaizuki, Eiheiji-cho, Yoshida-gun, Fukui 910-1193, Japan; 4Life Science Innovation Center, Faculty of Medical Sciences, University of Fukui, 23-3 Matsuokashimoaizuki, Eiheiji-cho, Yoshida-gun, Fukui 910-1193, Japan; 5Department of Neurology, Nakamura Hospital, 5-15 Tenno-cho, Echizen-city, Fukui 915-0068, Japan; n.miwako@ca3.so-net.ne.jp; 6Department of Radiology, Faculty of Medical Sciences, University of Fukui, 23-3 Matsuokashimoaizuki, Eiheiji-cho, Yoshida-gun, Fukui 910-1193, Japan; kimura@u-fukui.ac.jp; 7Sukoyaka Silver Hospital, 93-6 Shimadera-cho, Fukui 910-3623, Japan; t-suko@fukui-sukoyaka-silver.or.jp; 8Kimura Hospital, 57-25 Kitakanazu, Awara-city, Fukui 919-0634, Japan; y-kimura@kimura-hospital.jp; 9Ota Memorial Hospital, Fukuyama, Hiroshima 720-0825, Japan; kuriyama@shouwa.or.jp

**Keywords:** vitamin B_12_, homocysteine, cognitive impairment, MMSE, hippocampal atrophy, MRI-VSRAD

## Abstract

Vitamin B_12_ deficiency is associated with cognitive impairment, hyperhomocysteinemia, and hippocampal atrophy. However, the recovery of cognition with vitamin B_12_ supplementation remains controversial. Of the 1716 patients who visited our outpatient clinic for dementia, 83 had vitamin B_12_ deficiency. Among these, 39 patients (mean age, 80.1 ± 8.2 years) had undergone Mini-Mental State Examination (MMSE) and laboratory tests for vitamin B_12_, homocysteine (Hcy), and folic acid levels. The hippocampal volume was estimated using the z-score of the MRI-voxel-based specific regional analysis system for Alzheimer’s disease. This is multi-center, open-label, single-arm study. All the 39 patients were administered vitamin B_12_ and underwent reassessment to measure the retested for MMSE and Hcy after 21−133 days (median = 56 days, interquartile range (IQR) = 43–79 days). After vitamin B_12_ supplementation, the mean MMSE score improved significantly from 20.5 ± 6.4 to 22.9 ± 5.5 (*p* < 0.001). Hcy level decreased significantly from 22.9 ± 16.9 nmol/mL to 11.5 ± 3.9 nmol/mL (*p* < 0.001). Significant correlation was detected between the extent of change in MMSE scores and baseline Hcy values. The degree of MMSE score was not correlated with hippocampal atrophy assessed by the z-score. While several other factors should be considered, vitamin B_12_ supplementation resulted in improved cognitive function, at least in the short term, in patients with vitamin B_12_ deficiency.

## 1. Introduction

Cognitive impairment is a progressive disease that puts a heavy burden on patients, their families, and society in today’s aging society. Changes in the prevalence of dementia can alter social costs [1]. Most causes of cognitive impairment are not easy to treat, such as Alzheimer’s disease (AD), vascular dementia (VaD), dementia with Lewy body disease (DLB), and frontotemporal dementia. However, vitamin deficiencies, including folate or vitamin B_12_, are expected to inhibit the progression of cognitive impairment by vitamin administration [2,3] Although the prevalence of vitamin B_12_ shortage among all patients with dementia is not high [4,5], it should not be overlooked.

As humans cannot synthesize vitamin B_12_, it is essential to consume animal products that contain vitamin B_12_. Vitamin B_12_ deficiency is caused by impaired intake and absorption in the digestive tract. Vitamin B_12_ deficiency causes cognitive impairment, macrocytic anemia, and peripheral neuropathy. Hcy is converted to methionine by methionine synthase (Figure 1). Methionine synthase uses vitamin B_12_ as a cofactor. Vitamin B_12_ deficiency can cause hyperhomocysteinemia (HHcy). Folate is also a cofactor involved in the remethylation of Hcy [2,6]. Hcy is also transformed Hcy into cysteine by vitamin B_6_. Deficiency in folate and vitamin B_6_ can also cause HHcy. Vitamin B_12_ is not directly involved in nucleic acid synthesis; however, its deficiency results in the impairment of tetrahydrofolate synthesis, which in turn impairs nucleic acid synthesis [6,7,8].

Figure 1 shows the vitamin B_12_ and the homocysteine (Hcy) metabolic pathway. Vitamin B_12_ is a coenzyme involved in the remethylation of Hcy. Vitamin B_12_ deficiency causes hyperhomocysteinemia. Vitamin B_12_ deficiency also impairs nucleic acid synthesis.

Vitamin B_12_ deficiency can cause oxidative stress via the following mechanisms: vitamin B_12_ deficiency causes HHcy (Figure 1), and an increase in the Hcy level can lead to oxidative stress. Furthermore, vitamin B_12_ deficiency induces a reduction in reactive oxygen species (ROS) scavenging, indirectly induces a reduction in glutathione levels (Figure 1), and disturbs immune reaction regulation by altering the expression of cytokines and growth factors with consequent mild inflammatory reactions [9]. HHcy causes asymptomatic brain damage due to oxidative stress [8], atherosclerosis [7], and brain atrophy. HHcy is associated with AD [3,6,8,9,10,11,12,13,14], VaD [12,15,16], Parkinson’s disease [17,18], and stroke [12,19,20]. Low vitamin B_12_ levels have been written to be associated with HHcy, which in turn is related to cognitive impairment [21,22] and brain atrophy, including in the hippocampus [23]. However, the efficacy of vitamin B_12_ supplementation in cognitive recovery remains controversial. Some meta-analyses have reported that vitamin B_12_ improves cognitive impairment [24]. However, other clinical trial [25] and meta-analyses have reported that vitamin B_12_ did not improve cognitive impairment [26,27,28]. The association between Hcy reduction, B_12_ supplementation, and cognition is inconclusive. In the previous clinical trial and meta-analysis, the included participants were not limited to persons with only vitamin B_12_ deficiency. Furthermore, in the previous studies, not only was vitamin B_12_ administered but folate or vitamin B_6_ was also administrated. Therefore, we explored the effects of only vitamin B_12_ administration on cognitive impairment in patients with vitamin B_12_ deficiency and without folate deficiency. We also examined the correlation between hippocampal atrophy, cognitive decline, and Hcy levels.

## 2. Materials and Methods

### 2.1. Ethics, Approval, and Consent to Participate in This Research

This is multi-center, open-label, single-arm study. Patients seen at the outpatient dementia clinic at the University of Fukui Hospital or the Nakamura Hospital from January 2008 to December 2021 were included in the study. Medical history and oral medications, including vitamin supplementation, were also recorded. Clinical tests were performed for vitamins B_12_, B_1_, folic acid, Hcy, mean corpuscular volume (MCV), thyroid function, Mini-Mental State Examination (MMSE), and brain magnetic resonance imaging (MRI). The sensitive markers of vitamin B_12_ deficiency, including holotranscobalamin and methylmalonic acid, were not assessed in this study. Participants with low vitamin B_12_ levels (<172 pmol/L) were included in the study, and vitamin B_12_ (1500 μg/day) was administered via methylcobalamin tablets (Eisai Co., Ltd., Tokyo, Japan). The MMSE and serum Hcy level were reassessed after 21–133 days (median = 56 days, interquartile range (IQR) = 43–79 days) following the initiation of the vitamin B_12_ supplementation. A follow-up MMSE measurement was performed at 0.5, 1, or 2 years later (Appendix A).

This study was approved by the Institutional Review Board of the University of Fukui (20180092). All human materials were taken following the standards outlined in the Declaration of Helsinki principles of 1975, as revised in 2008 (http://www.wma.net/en/10ethics/10helsinki/ (accessed on 1 February 2022)).

### 2.2. Blood Tests

The ADVIA Centaur XP Immunoassay System (Siemens Healthcare Diagnostics Manufacturing Ltd., Dublin, Ireland) and its support equipment (Siemens Healthcare Diagnostics Inc. Diagnostics Incorporated, New York, NY, USA) were used to determine vitamin B_12_ and folate concentrations on the same day. The Hcy value was quantified using an atmospheric pressure ionization 3200 LC-MS/MS system (SCIEX, Tokyo, Japan) [2,28].

### 2.3. Scoring of Brain Atrophy by Z-Score of Voxel-Based Specific Regional Analysis System for Alzheimer’s Disease (VSRAD)

All standard MRI examinations were performed using a 1.5 T or 3.0 T GE Signa scanner at the University of Fukui Hospital or a 1.5 T GE-Healthcare Optima MR 360 at Nakamura Hospital.

VSRAD evaluates the relative regional brain volume of individual patients by statistically comparing the 3D T1-weighted images of the whole brain obtained at approximately 1 mm with a database of brain images of healthy elderly people using voxel-based morphometry. VSRAD is a free software widely used in Japan, and detailed information is provided in our previous study [2].

VSRAD applies statistical parametric mapping (SPM) to evaluate the regional volume of the brain of individual patients by statistically comparing it with a preloaded brain imaging database of 80 normal controls aged 54−86 years. After tissue segmentation and anatomical standardization, isotropic 8 mm cubic smoothing was performed to reduce individual differences in brain function localization, improve the signal-to-noise ratio, and make the count rate distribution of the image closer to the normal distribution. In the statistical test on the image database of healthy subjects, z-scores, which indicate how many standard deviations the gray matter volume and white matter volume of each patient are from the mean volume of healthy subjects, were calculated for each brain region by the average image and standard deviation image of the image database, and it appeared as a color scale map in the subject’s brain. The test range was predetermined. The test range was a fixed area that superimposed predetermined gray matter and white matter mask images. In the VSRAD advance, the target region of interest was determined in the medial temporal area, including the olfactory cortex, peach, and hippocampus, based on the results of the group analysis using SPM between patients with early AD and age-matched normal elderly patients. Atrophy (mean value of positive z-scores in the target area of interest) is the standard index in the VSRAD advance. A score of 0 to 1 indicates almost no atrophy, 1 to 2 indicates some atrophy, 2 to 3 indicates considerable atrophy, and ≥3 indicates strong atrophy [2,29,30].

### 2.4. Statistical Analysis

The data are uniformly presented as medians and IQRs. Comparisons of MMSE and Hcy levels before and after vitamin B_12_ supplementation were evaluated by the Wilcoxon’s signed-rank test because the data did not follow a normal distribution. Correlations were also evaluated using Pearson’s correlation coefficient if the data followed a normal distribution. If the data did not follow a normal distribution (vitamin B_12_ (pre), Hcy (pre), MMSE (post), and Hcy change), Spearman’s rank correlation coefficient was applied. Linear mixed models were applied to analyze the effects of confounding factors, such as age, sex, education, and follow-up interval between first examination and re-evaluation after starting vitamin B_12_. Missing values were handled using the list-wise deletion method. Statistical analyses were done utilizing IBM SPSS Statistics version 27 (IBM Corp, Armonk, NY, USA). *p* < 0.05 was considered statistically significant. By utilizing the free software G* Power 3.1, the power of the dataset was determined [31].

## 3. Results

### 3.1. Vitamin B_12_ Deficit and Hyperhomocysteinemia

Of the 1716 patients who had been examined in the outpatient clinic for dementia during the evaluation period, 83 had vitamin B_12_ deficiency. Of these, 44 were excluded (Figure 2). Hcy levels were not obtained in four patients. MMSE was not performed before treatment in three patients. Thirteen patients with concomitant folate deficiency [32] and 25 patients withdrew from the examination. Finally, 39 patients (22 males and 17 females) were incorporated in this research (Table 1). The mean age of the participants was 80.3 ± 8.2 years. The median educational status (years) was 12 years (IQR 8–12); 26 patients underwent VSRAD, and 34 underwent MCV. A comparison of the 39 patients with 44 excluded patients is shown in Appendix A. Among the subjects who had been examined in our outpatient clinic, vitamin B_12_ deficiency was 4.8% (83/1716 patients). The median vitamin B_12_ concentration was 142 pmol/L (IQR 122–154) (normal range 172–674) [33]. The median MMSE before treatment (baseline) was 22 (IQR 15–26), median Hcy was 16.7 mmol/mL (IQR 12.0–27.7) (normal range 3.7–13.5), and median folate was 6.8 pg/mL (IQR 5.0–8.4) (normal range 3.6−12.9) (Table 1). Detailed data for each of the 39 patients are presented in Appendix A. A significant inverse correlation was observed between baseline Hcy and vitamin B_12_ levels (ρ = −0.416, *p* = 0.008) (Figure 3).

Inverse correlation was shown significantly between baseline vitamin B_12_ and Hcy levels (ρ = −0.416, *p* = 0.008). The Spearman rank correlation coefficient (ρ) was applied as the data did not follow the normal distribution.

### 3.2. Hippocampal Atrophy and Hyperhomocysteinemia

The median z-score correlated with hippocampal volume in 24 patients was 1.7 (IQR 1.2–2.6), indicating mild atrophy [29] (Table 1). No significant correlation was observed between the z-score and baseline Hcy level (ρ = 0.058, *p* = 0.792) (Figure 4A). No significant correlation was detected between the z-score and baseline vitamin B_12_ levels (ρ = −0.041, *p* = 0.853) (Figure 4B). However, a significant correlation was seen between hippocampal atrophy assessed by the z-score and the baseline MMSE score (ρ = −0.445, *p* = 0.033) (Appendix A).

### 3.3. Changes in Homocysteine Levels and MMSE Score by Vitamin B_12_ Supplementation

Supplementation with vitamin B_12_ changed the mean vitamin B_12_ level from 135.8 ± 27.5 pmol/L to 721.0 ± 523.1 pmol/L (*p* < 0.001). The average Hcy level decreased significantly from 22.9 ± 16.9 to 11.5 ± 3.9 nmol/mL in a short period of time (*p* < 0.001) (Figure 5). No correlation was observed between the degree of decrease in the Hcy level and that of change in the vitamin B_12_ level (ρ = 0.225, *p* = 0.240) (Appendix A). The MMSE score improved significantly during a short period from 20.5 ± 6.4 days to 22.9 ± 5.5 days following the vitamin B_12_ administration (*p* < 0.001; effect size, r = 0.73) (Figure 6). The power of the data following the vitamin B_12_ supplementation was 0.955. Confounders, age (*p* = 0.152), sex (*p* = 0.421), and education (*p* = 0.267) had no effects on MMSE change. The confounder follow-up interval between starting vitamin B_12_ supplementation and MMSE reevaluation (*p* = 0.448), and the total follow-up period (*p* = 0.614) also had no effect on the change in the MMSE score. The confounder follow-up interval between the initiation of vitamin B_12_ supplementation and the reevaluation of the Hcy level (*p* = 0.578) had no effect on the reduction in the Hcy level. There was no correlation between baseline MMSE scores and baseline Hcy levels (Appendix A). A significant correlation was seen between the change in MMSE and the baseline Hcy level (ρ = 0.318, *p* = 0.049) (Figure 7A), while no correlation was found between the change in MMSE and the change in Hcy with vitamin B_12_ supplementation (Figure 7B). If the patients who exhibited a baseline Hcy level of >50 nmol/mL were removed, the change in the MMSE and the baseline Hcy level would exhibit no correlation (ρ = 0.257, *p* = 0.136). Furthermore, if the patients who exhibited a change in the Hcy level of >40 nmol/mL from the baseline level were removed, the change in the MMSE and the degree of decrease in the Hcy level would exhibit no correlation (ρ = 0.120, *p* = 0.551).

The Wilcoxon signed-rank test was applied because the date of Hcy before treatment did not followed normal distribution.

A significant correlation was found between MMSE changes and baseline MMSE scores (ρ = −0.603, *p* < 0.0001) (Appendix A). No correlation was observed between the change in MMSE and the z-score (Figure 8). Long-periods follow-up of the MMSE was performed at 0.5, 1, and 2 years, and analyzed using the GLMM. Even though the MMSE score declined despite continuous vitamin B_12_ supplementation, the decrease was gradual, as demonstrated in Appendix A.

### 3.4. Vitamin B_12_ Deficiency and MCV

In this study, MCV was slightly higher, median 97.0 fL (IQR 91.0–97.9) (normal range 83.6−98.2 fL) [2]. Nevertheless, no correlation was observed between vitamin B_12_ levels and MCV (ρ = −0.125, *p* = 0.487) (Appendix A).

## 4. Discussion

In this multi-center, open-label, single-arm study, it was evidenced that (1) vitamin B_12_ deficiency among patients who visited the outpatient dementia clinic was 4.8%; (2) Low vitamin B_12_ levels were related to HHcy, and vitamin B_12_ supplementation improved HHcy in the short-term; (3) Vitamin B_12_ supplementation improved the MMSE score in the short term. A significant correlation was found between the degree of improvement in MMSE and baseline Hcy; (4) A significant correlation was detected between hippocampal atrophy and baseline MMSE score. Significant correlation was not recognized between hippocampal atrophy and baseline Hcy levels or the degree of change in MMSE scores.

In this study, patients with isolated vitamin B_12_ deficiency accounted for 5.0% of all patients with dementia. Previous reports have shown that the prevalence of vitamin B_12_ among patients with dementia is 7.5% [34]. Another report has shown that vitamin B_12_ shortage affects approximately 5% of people aged 65–74 years and more than 10% of people aged ≥75 years [35].

Low vitamin B_12_ levels were clearly related to HHcy, and vitamin B_12_ supplementation improved HHcy in this study. These results are consistent with those of previous findings [3,15,36]. HHcy is related to cognitive impairment [11,12] and is positively correlated with brain atrophy [3,23,37,38,39,40]. The possible mechanisms of Hcy-induced cognitive impairment include HHcy, which induces cell death by DNA and oxidative damage; it is also associated with amyloid β protein and tau [6,12,17,41]. This causes NO-mediated dysfunction of the vascular endothelium and amyloid angiopathy, leading to vascular damage [41,42]. In this study, we observed a rapid improvement in the Hcy level following the vitamin B_12_ supplementation. Vitamin B_12_ supplementation and Hcy reduction may be associated with improved neuronal death [6].

Vitamin B_12_ supplementation has been associated with a slower decline in cognitive and clinical functions, as shown in previous reports [3,43,44]. However, it is not clear whether vitamin B_12_ supplementation alone improves cognitive function over a short period of time. In this study, vitamin B_12_ supplementation improved the MMSE score, at least in the short term. One possible reason for the improvement of MMSE score with vitamin B_12_ supplementation is the improvement in attention and mood [24]. Vitamin B_12_ supplementation has also been reported to reduce the symptoms of depression [45], fatigue, delirium [46] and abnormalities in electroencephalography [47]. Other possible explanations include a practice effect due to the possibility of remembering the content of the test and a placebo effect. Another possible reason is that doctors and healthcare providers provided guidance on how to improve lifestyles to activate brain function, such as encouraging a regular lifestyle, diet, and participation in social activities at the dementia outpatient clinic. The question of whether an improvement in the MMSE of <3 points is clinically significant remains. First, it was described that the MMSE has a specificity and sensitivity for predicting dementia of 82% and 87%, respectively [48]. In a previous randomized placebo-controlled trial of donepezil in patients with AD, the following conclusions were made. Improvements in the MMSE favoring donepezil were exhibited at weeks 6, 12, and 24 and at the end of the follow-up period. The improvements in the MMSE from the baseline value at weeks 6, 12, and 24 following the initiation of treatment were 1.4 (*p* = 0.02), 1.2 (*p* = 0.03), and 1.8 (*p* = 0.002) points, respectively. In that study, the placebo group exhibited an improvement in the MMSE of <0.3 points [49]. Donepezil is one of the gold-standard drugs used for treating AD. In this study, the improvement in the MMSE score exhibited following the treatment with vitamin B_12_ was 2.5 points (*p* < 0.0001; effect size, r = 0.73). Therefore, the change in the MMSE from vitamin B_12_ treatment can be significant.

The extent of MMSE score recovery was correlated with baseline Hcy concentrations (Figure 7A). However, if the patients who exhibited a baseline Hcy level of >50 nmol/mL were removed, the change in the MMSE and the baseline Hcy level would exhibit no correlation. The extent of improvement of the MMSE score was not correlated significantly with that of the VSRAD z-score (Figure 8). So, even with serious hippocampal atrophy, vitamin B_12_ administration is expected to improve cognitive function.

In the GLMM, the decrease in MMSE score was slow in vitamin B_12_ deficiency patients with vitamin B_12_ supplementation in this study (Appendix A). Vitamin B_12_ supplementation has been written to be related to slow progress of brain atrophy [23,50], particularly in the hippocampus, medial parietal lobe, and occipital cortex of patients with mild cognitive impairment patients [23]. Therefore, vitamin B_12_ treatment can decrease the decline of the progression of cognitive function.

In this study, hippocampal atrophy was not associated with baseline levels of Hcy or vitamin B_12_. Hcy levels have been known to be positively correlated with cerebral atrophy [3], cortical [23,37,39], subcortical [38,39], and hippocampal atrophy [39,40]. HHcy patients treated with B vitamins have been reported to have lower Hcy, reduced gray matter atrophy, and slow cognitive decline [51]. One possible explanation is that the degree of HHcy was not correlated with hippocampal atrophy because the presence of other pathologies (AD, VaD, DLB, etc.) influenced the degree of hippocampal atrophy.

In this study, while MCV values did not correlate with B_12_ levels, MCV was generally higher (median 97.0 fL (IQR 91.0–97.9) (Normal range: 83.6–98.2) [2]. These findings agree with previous reports of B_12_ or folate shortage [2,46,51]. Vitamin B_12_ and folate are not included in routine blood tests; however, MCV is a routine test. Therefore, if MCV is high, the possibility of vitamin B_12_ or folate shortage should be considered in patients with dementia.

This study had some limitations. First of all, the participant number was small, with no control group, and not all the participants followed up long enough. Second, we did not examine the depression scale, Geriatric Depression Scale 15. We also did not measure vitamin B_6_, which is responsible for HHcy. The MRI-VSRAD z-score was measured using three different scanners. We estimated the cognitive function of patients by using only the MMSE, which has a specificity and sensitivity for predicting dementia of 82% and 87%, respectively [48]. The degree of cognitive impairment was relatively mild in this study (mean MMSE, 20.4 ± 6.7). Because we provided guidance on how to improve lifestyle, without a randomized study of patients who were not vitamin B_12_ deficient in a similar situation, we cannot assume that vitamin B_12_ deficiency improved the MMSE score and not because of other care provided to patients. A more systematic trial that includes all the details missing from this pilot study is suggested.

## 5. Conclusions

In patients with vitamin B_12_ deficiency, vitamin B_12_ supplementation resulted in a reduction of Hcy and improved cognitive function, at least in the short period, irrespective of hippocampal atrophy. The extent of improvement in the MMSE score after vitamin B_12_ supplementation was correlated with the baseline Hcy value. However, the improvement of cognitive function involves several factors, including practice effects, placebo effects, and mood improvement by vitamin B_12_ supplementation.

## Figures and Tables

**Figure 1 nutrients-14-01494-f001:**
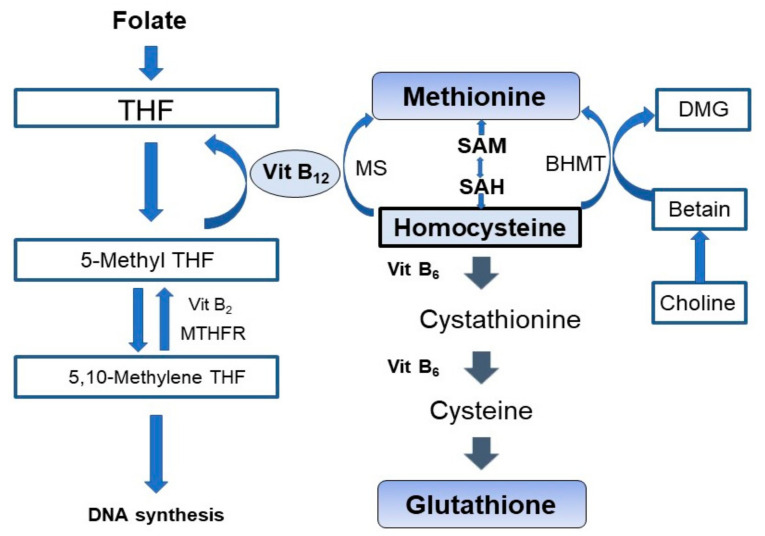
Schematic representation of Homocysteine (Hcy) metabolism. Vitamin B_12_ and folate are cofactors in one-carbon metabolism, and they facilitate Hcy remethylation. Vitamin B_12_ and folate shortage prevents the conversion of Hcy to methionine and leads hyperhomocysteinemia (HHcy). Vitamin B_6_ shortage prevents the transfer of Hcy to cystathionine and induce HHcy, too. THF, tetrahydrofolate; Vit B_12_, vitamin B_12_; 5-Methyl THF, 5-methyltetrahydrofolate; Vit B_2_, vitamin B_2_; MTHFR, 5, 10-methylenetetrahydrofolate reductase; 5, 10-Methylene THF, 5, 10-methylenetetrahydrofolate; MS, methionine synthase; SAM, S-adenosylmethionine; SAH, S-adenosylhomocysteine; Vit B_6_, vitamin B_6_; BHMT, betaine Hcy methyltransferase; DMG, dimethylglycine.

**Figure 2 nutrients-14-01494-f002:**
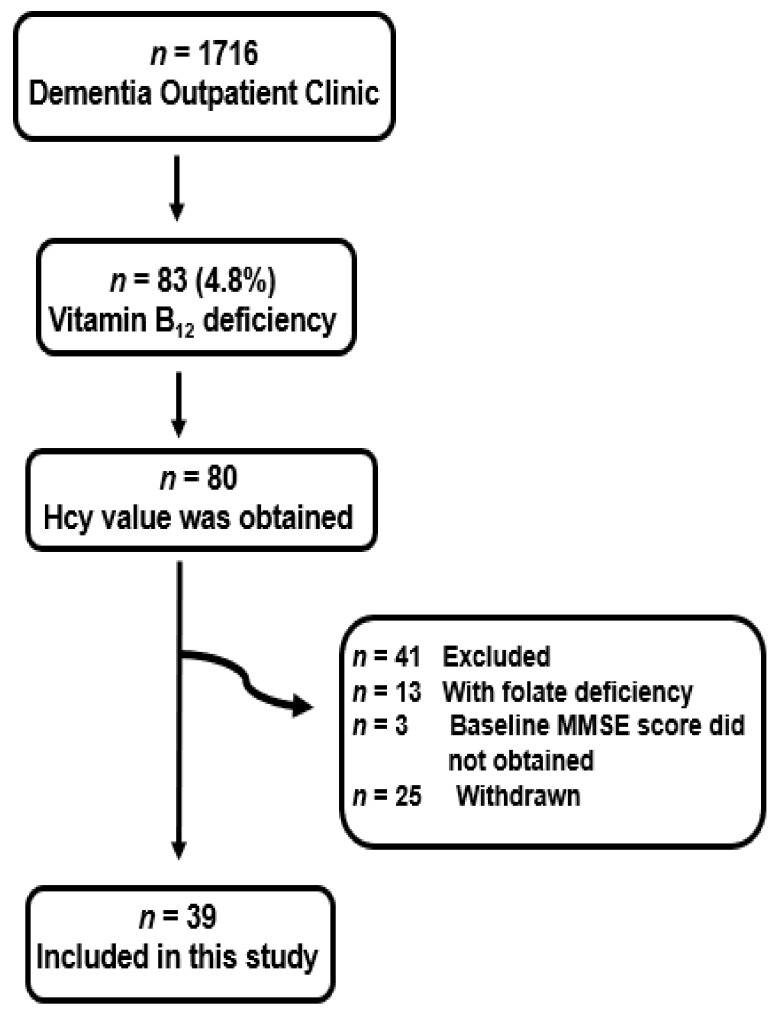
Flow chart describing how to select the research participants. Among them, homocysteine (Hcy) is determined in 80 patients. Of these, 41 patients are omitted for the causes explained as follows: concomitant folate shortage (*n* = 15), Mini-Mental State Examination (MMSE) before treatment was not performed (*n* = 3), or withdrawal from the study (*n* = 25). In total, 39 patients are enrolled in this study.

**Figure 3 nutrients-14-01494-f003:**
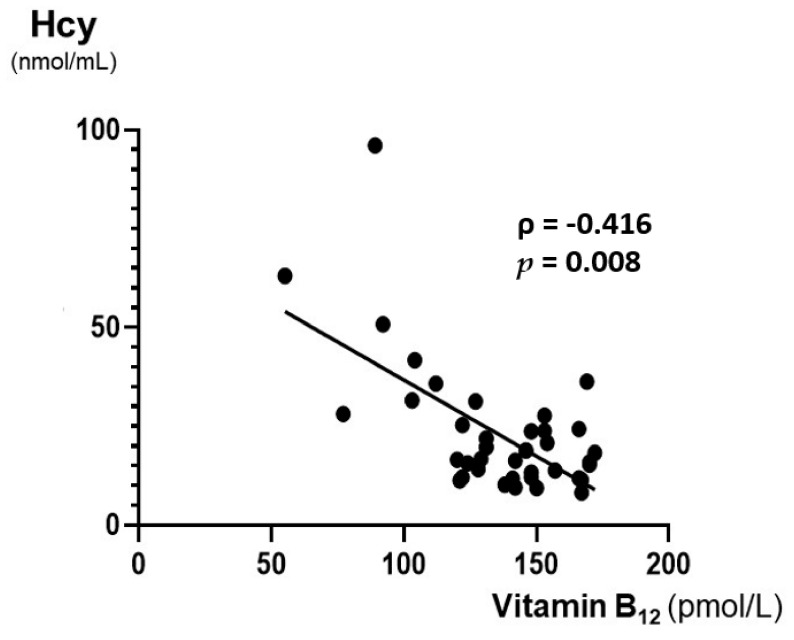
Plasma vitamin B_12_ and homocysteine levels are inversely correlated.

**Figure 4 nutrients-14-01494-f004:**
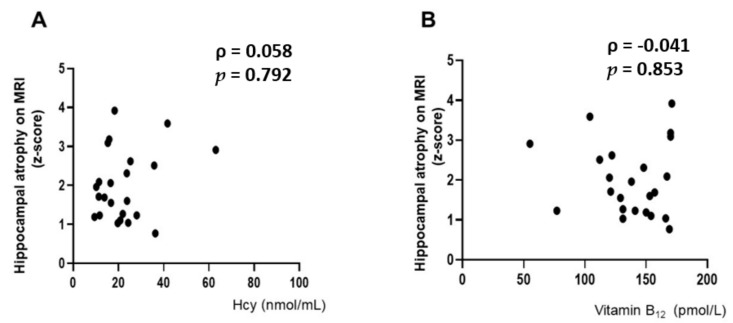
No significant correlation is shown between the z-score and baseline homocysteine (Hcy) (ρ = 0.058, *p* = 0.792) (**A**). No significant correlation is found between the z-score and baseline vitamin B_12_ (ρ = −0.041, *p* = 0.853) (**B**). Spearman’s rank correlation coefficient (ρ) was applied as the data for Hcy and vitamin B_12_ deviated from the normal distribution.

**Figure 5 nutrients-14-01494-f005:**
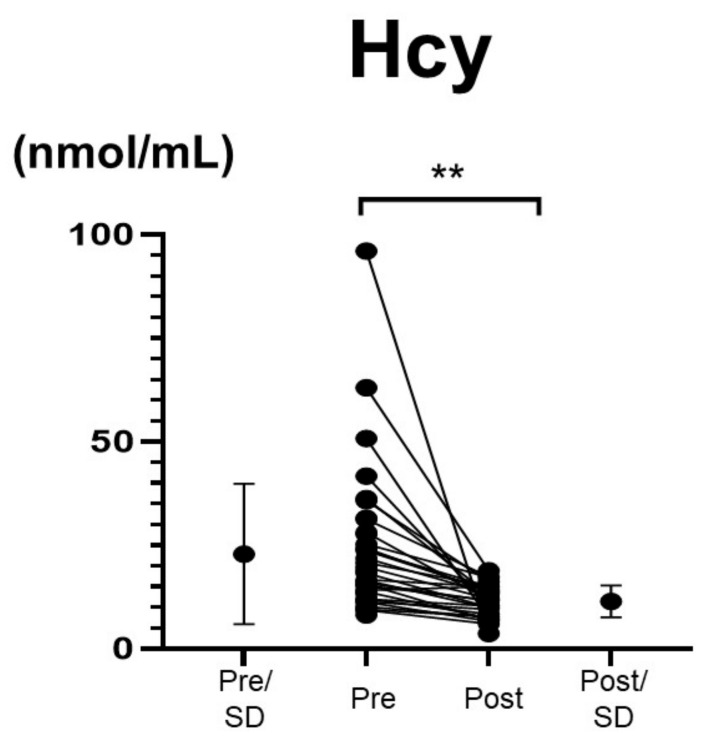
Vitamin B_12_ supplementation decreased plasma homocysteine (Hcy) concentrations. Vitamin B_12_ administration significantly decreased Hcy concentrations from 22.9 ± 16.9 to 11.5 ± 3.9 nmol/mL (** *p* < 0.0001).

**Figure 6 nutrients-14-01494-f006:**
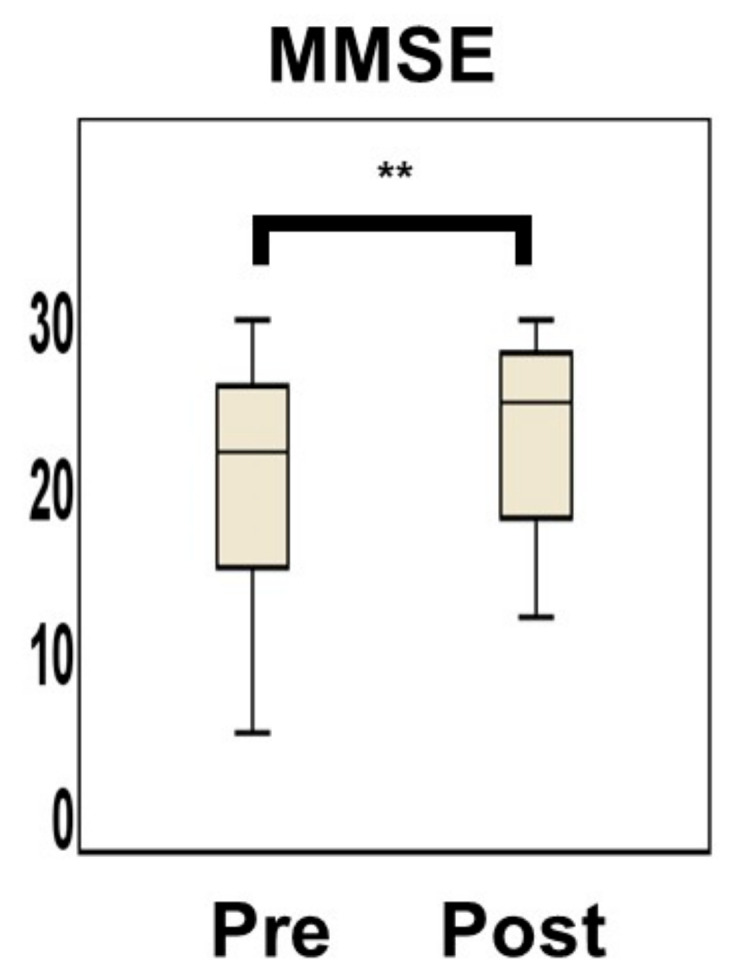
Mini-Mental State Examination (MMSE) score improved significantly from 20.5 ± 6.4 to 22.9 ± 5.5 (** *p* < 0.0001, effect size r = 0.73), 21−133 days after vitamin B_12_ supplementation. Bar ± SD. The Wilcoxon signed-rank test was applied as the MMSE score after treatment deviated from the normal distribution.

**Figure 7 nutrients-14-01494-f007:**
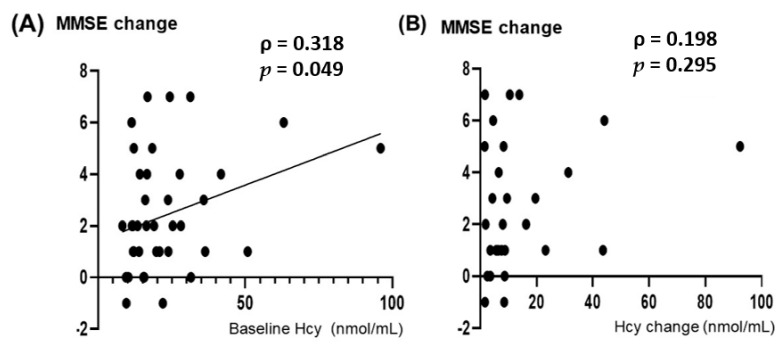
A significant correlation is demonstrated between the degree of change in the Mini-Mental State Examination (MMSE) and the baseline Hcy value (ρ = 0.318, *p* = 0.049) (**A**). However, the degree of Hcy change and MMSE change did not correlate significantly (ρ = 0.198, *p* = 0.295) (**B**). Spearman’s rank correlation coefficient (ρ) was applied as Hcy and the degree of Hcy change deviated from the normal distribution.

**Figure 8 nutrients-14-01494-f008:**
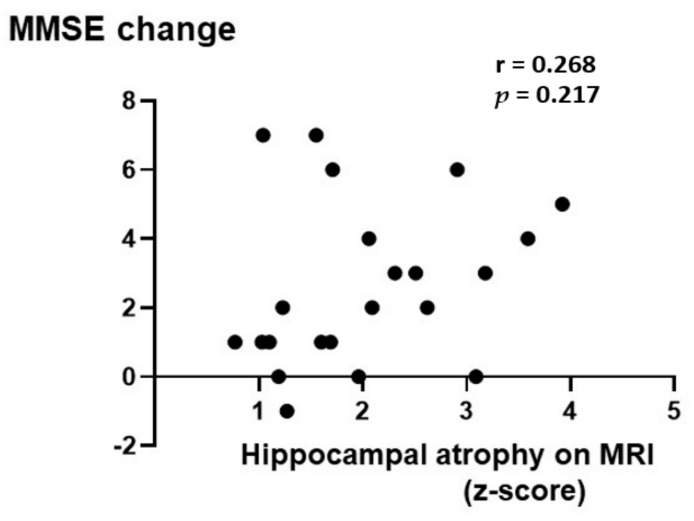
No correlation is shown between the change in Mini-Mental State Examination (MMSE) and z-score (r = 0.268, *p* = 0.217). The change in MMSE score and z-score with normal distribution was analyzed using Pearson’s correlation coefficiency (r).

**Table 1 nutrients-14-01494-t001:** Demographic data of 39 patients with B_12_ deficiency.

Age (Mean ± SD)	80.1 ± 8.2
Range	49–91
Male sex, *n* (%)	22 (56.4)
Education (Year) (Median (IQR))	12 (8–12)
Range	6–16
MMSE (Median (IQR))	22 (15–26)
Range	5–30
Vitamin B_12_ (Median (IQR)), pmol/L	142 (122–154)
Range	55–171
Normal Range	(172–674) [33]
Folate (Median (IQR)), ng/mL	6.8 (5.0–8.4)
Range	3.8–53.0
Normal Range	(3.6–12.9) [2]
Hcy(Median (IQR)), nmol/mL	16.7 (12.0–27.7)
Range	8.2–96.0
Normal range	(3.7–13.5) [2]
MCV (Median (IQR)), fL	97.0 (91.0–97.9)
Range	81.2–108.0
Normal range	(83.6–98.2) [2]
MRI hippocampal atrophy	
Z-score (Median (IQR))	1.7 (1.2–2.6)
Range	0.8–3.1
Cutoff	1.35 [29]

Abbreviations: SD, standard deviation; IQR, interquartile range; MMSE, Mini-Mental State Examination; Hcy, homocysteine; (), normal range; MCV, mean corpuscular volume; MRI, magnetic resonance imaging.

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
