# Peer review of "Influences of Vitamin B12 Supplementation on Cognition and Homocysteine in Patients with Vitamin B12 Deficiency and Cognitive Impairment"

_nutrients, 2022, doi:10.3390/nu14071494_

Round 1
Reviewer 1 Report
This is an interesting paper, reporting on the contribution of cobalamin (vit. B12) deficiency on homocysteine levels in plasma and cognitive functions in elderly psychatric patients. Although its a meritful waork, there are several flaws and corrections to be made.
Major:
- L. 34: Why was there such a high range? Why were no follow-ups made, with a longer period in the short-term patients?
- The authors should clearly define from the beginning that they only focuss on vitamin B12, as they excluded folate deficiency, although this may be an important contributor. They also did not include other importants nutrients, like vitamin B6 and choline/methyl donors in their study. The binary approach (B12 sufficient/insufficient) is in general clinically questionable, but in a study it can be justified; what they did, as folate insufficient patients were excluded here. It just requires detailed explanation.
- As the time between intiation of supplementation was very different, is there any data about the slope of B12-increase / HCy decrease?
- L. 56f and Figure 1: This is incomplete. 50% of methionine synthesis from homocysteine occurs via betaine:homocysteine methyl transferase (BHMT), depending on betaine, a downstream metabolite of choline. Choline is a critical nutrient like B12. Moreover, choline is important for the methionine synthase reaction as well: further downstream products of choline like dimethylglycine are 1-carbon donors of methyltetrahydrofolate formation that is required in the B12-dependent methionine synthase reaction. In general, choline/betaine are the methyl donors, whereas tertahydrofolate is methyl-carrier and B12 co-factor in methyl transfer for methionine synthase. In this context, figure 1 is essentially incomplete!
- L. 75: Specify, how oxidative stress is associated with B12, e.g. via glutathione formation according to figure 1.
- L 81ff: The reviewer is not surprized that in a nutrient situation with frequent mutual deficiency of B6, folate and B12, B12 alone results in inconclusive results. The more important is here to explain why and how this was approached anew in this study.
- Provide data on internationally established normal and cut-off values for plasma concentrations of B12, in pmolar units as well! Explain the B12 test? Why were sensitive markers of vit. B12 deficiency, holotranscobalamin and methylmalonic acid, not determined? Aat least for borderline values?
- L 140ff: Statistics. Define, how data are presented. Use medians and IQR uniformly, as normal distribution is not always guranteed.
- Table 1: Provide mediansns, IQR and full ranges. Provide cut-off values for deficiencies with references.
Minor:
- L 33: Insert number of patients.
- L. 34: Provide full range/medians+IQR. Data do not appear to be normally distributed.
- L.46f: Wording. Only >changes in prevalence< can alter social costs.
- L. 95f: Remove repetitive dosing.
- L. 97f: provide data as medians and IQR. The impression arises that patients were treted for very different durations. Or was the only difference that in follow-up visits? Explain, please!
- L 153-156: It should be clarified by the initial concept so that it is clear, why these patients were excluded.
- L 162: Provide reference for normal values.
- L. 182: The normal symbol is a greek r (Rho)".
- Fig. 3: Use molar units. The axes should have intercepts.
- Fig. 5: Provide data points and connecting lines for individual changes.
- L 242: >Proof< isn't the right term here. It is >scientific evidence<.
- L 246: improvement rather than change.
- L 250 insert: with >isolated< vitamin B12 deficiency.
- L 262: short term improvement: Is really meant here that the effect was only short-term? Isn't it meant that the effect was rapid and measured after only a short supplementation period?
- L 267: insert: vitamin B12 supplementation >alone< improves...
- L 280: >improve< rather than >restore<.
- L 286: >decrease the decline< rather than >improve the progression<.
- L 295: higher compared to what? Provide normal values and ref!
- L. 304-307: Rephrease positively, i. e. suggesting a more systematic trial, where all the missing details are included. based on this pilot study.
Author Response
Reviewer1.
Thank you for your time and effort in reviewing our manuscript. We appreciate your helpful comments and believe that they have significantly improved the quality of our manuscript. We have revised the manuscript as outlined below.
Major:
- L. 34: Why was there such a high range? Why were no follow-ups made, with a longer period in the short-term patients?
-> Thank you very much for your comments. Actually, in this study, the re-revaluation period of the MMSE score varied according to the patients’ individual circumstances. The follow-up period, with a median of 24 (IQR = 13.1) is described in Supplemental Table 1.
- The authors should clearly define from the beginning that they only focus on vitamin B12, as they excluded folate deficiency, although this may be an important contributor. They also did not include other important nutrients, like vitamin B6 and choline/methyl donors in their study. The binary approach (B12 sufficient/insufficient) is in general clinically questionable, but in a study, it can be justified; what they did, as folate insufficient patients were excluded here. It just requires detailed explanation.
-> We have revised the manuscript as follows: “Therefore, we explored the effects of the administration of only vitamin B12 on cognitive impairment in patients with vitamin B12 deficiency not accompanied by folate deficiency.” (L93–95)
- As the time between initiation of supplementation was very different, is there any data about the slope of B12-increase / Hcy decrease?
-> We have reassessed the relationship between the vitamin B12 increase and Hcy decrease. We have added the following description.
“No correlation was observed between the degree of decrease in the Hcy level and that of change in the vitamin B12 level (ρ=0.225, P=0.240) (Supplementary Figure S2).” (L217–218)
- L. 56f and Figure 1: This is incomplete. 50% of methionine synthesis from homocysteine occurs via betaine: homocysteine methyl transferase (BHMT), depending on betaine, a downstream metabolite of choline. Choline is a critical nutrient like B12. Moreover, choline is important for the methionine synthase reaction as well: further downstream products of choline like dimethylglycine are 1-carbon donors of methyltetrahydrofolate formation that is required in the B12-dependent methionine synthase reaction. In general, choline/betaine are the methyl donors, whereas tetrahydrofolate is methyl-carrier and B12 co-factor in methyl transfer for methionine synthase. In this context, figure 1 is essentially incomplete!
-> We have revised Figure 1 in accordance with your suggestions (Figure 1).
- L. 75: Specify, how oxidative stress is associated with B12, e.g., via glutathione formation according to figure 1.
-> We have added the following description of how vitamin B12 is associated with oxidative stress: “Vitamin B12 deficiency can cause oxidative stress via the following mechanisms: vitamin B12 deficiency causes HHcy (Figure 1), and an increase in the Hcy level can lead to oxidative stress. Furthermore, vitamin B12 deficiency induces a reduction in reactive oxygen species (ROS) scavenging, indirectly induces a reduction in glutathione levels (Figure 1), and disturbs immune reaction regulation by altering the expression of cytokines and growth factors with consequent mild inflammatory reactions [9].” (L76–81)
- L 81ff: The reviewer is not surprised that in a nutrient situation with frequent mutual deficiency of B6, folate and B12, B12 alone results in inconclusive results. The more important is here to explain why and how this was approached anew in this study.
-> As you suggest, this study included patients with only vitamin B12-deficiency, as those with folate deficiency were excluded. Likewise, vitamin B6 was not measured in this study.
We have revised the manuscript as follows:
“In the previous clinical trial and meta-analysis, the included participants were not limited to persons with only vitamin B12 deficiency. Furthermore, in the previous studies, not only was vitamin B12 administered but folate or vitamin B6 was also administrated. Therefore, we explored the effects of only vitamin B12 administration on cognitive impairment in patients with vitamin B12 deficiency and without folate deficiency.” (L90–95)
- Provide data on internationally established normal and cut-off values for plasma concentrations of B12, in pmolar units as well! Explain the B12 test? Why were sensitive markers of vit? B12 deficiency, holotranscobalamin and methylmalonic acid, not determined? At least for borderline values?
->We completely agree with your suggestion. We have added the pmolar unit in L107, L177, Table 1, Figure 3, Figure 4, L216, L217, Supplementary Figure S2, Supplementary Figure S6, Supplementary Table 1, and Supplementary Table 2.
We recognize the importance of the sensitive markers of vitamin B12 deficiency, holotranscobalamin and methylmalonic acid. However, in this study, these examinations were not performed.
We have added the following description: “The sensitive markers of vitamin B12 deficiency, including holotranscobalamin and methylmalonic acid, were not assessed in this study.” (L105, 106)
- L 140ff: Statistics. Define, how data are presented. Use medians and IQR uniformly, as normal distribution is not always guaranteed.
Thank you for your comment. We have added the following description: “The data are uniformly presented as medians and IQRs.” (L152)
- Table 1: Provide medians, IQR and full ranges. Provide cut-off values for deficiencies with references.
Thank you for your comment. We have added the IQR, full range, and cutoff value for the deficiencies, and have included the appropriate references (Table 1).
Minor:
1.L 33: Insert number of patients.
-> We have revised the manuscript as follows: “All 39 patients were administered vitamin B12 and underwent reassessment to measure the MMSE and Hcy after 21-133 days (median=56 days, IQR=36 days).” (L34, 35)
2L. 34: Provide full range/medians. Data do not appear to be normally distributed.
->
We have revised the manuscript as follows: “All the 39 patients were administered vitamin B12 and underwent reassessment to measure the retested for MMSE and Hcy after 21-133 days (median=56 days, IQR=36 days).” (L34, 35)
3L.46f: Wording. Only >changes in prevalence< can alter social costs.
Thank you very much for your comments.
->We have changed the description to the following: “Changes in the prevalence of dementia can alter social costs [1].” (L48, 49)
4L. 95f: Remove repetitive dosing.
-> We have removed the repetitive dosing description and revised the text as follows: “Participants with low vitamin B12 levels (<172 pmol/L) were included in the study, and vitamin B12 (1,500 μg/day) was administered via methylcobalamin tablets (Eisai Co., Ltd.).” (L107–109)
5L. 97f: provide data as medians and IQR. The impression arises that patient were treated for very different durations. Or was the only difference that in follow-up visits? Explain, please!
->We have added the median and IQR.
Per your kind suggestion, the follow-up visit was different for each patient. We have revised the manuscript to the following: “The MMSE and serum Hcy level were reassessed after 21–133 days (median=56 days, IQR=36 days) following the initiation of the vitamin B12 supplementation.” (L109–111)
6L 153-156: It should be clarified by the initial concept so that it is clear, why these patients were excluded.
-> In the introduction section, we have explained the reason why these patients were excluded. We have revised this description to the following: “In the previous clinical trial and meta-analysis, the included participants were not limited to persons with only vitamin B12 deficiency. Furthermore, in the previous studies, not only was vitamin B12 administered but folate or vitamin B6 was also administrated. Therefore, we explored the effects of only vitamin B12 administration on cognitive impairment in patients with vitamin B12 deficiency and without folate deficiency.” (L90–95)
7L 162: Provide reference for normal values.
->We have added a reference for the normal values (L176).
8L. 182: The normal symbol is a Greek r (Rho)".
- We have changed all of the R values to ρ. (L198, 203, 206, 211, 212, 213, 220, 230, 234, 235, 250, 251, 255, and 267).
9Fig. 3: Use molar units. The axes should have intercepts.
- We have used molar units (Figure 3).
10Fig. 5: Provide data points and connecting lines for individual changes.
- We have provided points and connecting lines for the individual changes (Figure 5).
11L 242: >Proof< isn't the right term here. It is >scientific evidence<.
- We have changed the description according to your suggestion (L267).
12L 246: improvement rather than change.
- We have revised the word “change” to “improvement.” (L271)
13L 250 insert: with >isolated< vitamin B12 deficiency.
- We have added the term “isolated.” (L275)
14L 262: short term improvement: Is really meant here that the effect was only short-term? Isn't it meant that the effect was rapid and measured after only a short supplementation period?
As you suggest, the effect was rapid, and it was measured after only a short supplementation period. So, we have changed the description to the following: “In this study, we observed a rapid improvement in the Hcy level following the vitamin B12 supplementation.” (L287, 288)
15L 267: insert: vitamin B12 supplementation >alone< improves...
-> We have inserted the term “alone.” (L292)
16L 280: >improve< rather than >restore<.
-> We have changed the term used in this description from “restore” to “improve.” (L315)
17L 286: >decrease the decline< rather than >improve the progression<.
-> We have changed the description from “improve the progression” to “decrease the decline.” (L320-321)
18L 295: higher compared to what? Provide normal values and ref!
- We have added the normal value for the MCV and added the appropriate reference [2] (L330).
19 L. 304-307: Rephrase positively, i. e. suggesting a more systematic trial, where all the missing details are included. based on this pilot study.
- We have added the following description: “A more systematic trial that includes all the details missing from this pilot study is suggested.” (L343–344)
Reviewer 2 Report
This manuscript reports data demonstrating that vitamin B12 supplementation results in improved MMSE scores in B12-deficient patients visiting an outpatient clinic for dementia. This is an important study given the aging population and the prevalence of B12 deficiency in older adults. However, some concerns should be addressed.
Major concerns:
- There seem to be huge variations in both the enrollment in the study (a period of 13 years) and in the length of B12 supplementation (21-133 days). Confounding by either of these does not seem to be addressed.
- The central finding of the manuscript is that Hcy is lowered post supplementation (Figure 5) and that MMSE is increased post supplementation (Figure 6). The real effect of B12 supplementation is somewhat hard to interpret given that all subjects were assessed after different lengths of supplementation.
- Perhaps most importantly, is an increase in of <3 in MMSE score a clinically meaningful change? This should be addressed. Similarly, the authors do not address the lack of specificity that is inherent in the MMSE.
- The application of the generalized mixed model is not clear. Were follow up MMSE measures taken at 0.5, 1, or 2 years? Is the GLMM used to predict this? This needs clarification in the methods section.
- The authors state in the discussion that this is a pre/post study without a control group. This should be stated in the abstract and/or introduction for full transparency.
Minor concerns:
- On page 2 (lines 53-63) the authors use “folic acid” in many places were “folate would be more appropriate.
- Also on page 2 (lines 53-63), the authors state that B12 deficiency causes failure to convert dUMP to dTMP, which is true. However, it can also starve the cell of other nucleotide cofactors such as purines. Maybe more correct to just leave this as “tetrahydrofolate synthesis is impaired, which in turn impairs nucleic acid synthesis.”
- Page 2 (lines 75-76): probably more appropriate to say HHcy “is associated with” rather than “lead to” AD, VaD, etc.
- Text says 1716 patients enrolled, figure 2 says 1715.
- Page 4 (lines 155-156): sentence starting “with folic acid” is not a complete sentence.
- Figure 7: In both panels the trend lines seem to be driven by a few outliers with exceedingly high Hcy levels. This should be addressed.
Author Response
Reviwer2.
Thank you for your time and effort in reviewing our manuscript. We appreciate your helpful comments and feel that they have significantly improved the quality of our manuscript. We revised the manuscript as outlined below.
Major concerns:
There seem to be huge variations in both the enrollment in the study (a period of 13 years) and in the length of B12 supplementation (21-133 days). Confounding by either of these does not seem to be addressed.
- Thank you very much for your comments. As you suggest, the interval between the initiation of vitamin B12 supplementation and the reevaluation of the MMSE and Hcy level ranged from 21–133 days. However, the total supplementation period exhibited a median of 24 months and an IQR of 13.1 months (Supplemental Table S1).
We had added the following description: “The confounder follow-up interval between starting vitamin B12 supplementation and MMSE reevaluation (P = 0.448), and the total follow-up period (P = 0.614) also had no effect on the change in the MMSE score. The confounder follow-up interval between the initiation of vitamin B12 supplementation and the reevaluation of the Hcy level (P = 0.578) had no effect on the reduction in the Hcy level” (L222–226). The confounding variable of when the participant was enrolled (P = 0.352) had no effect on the change in the MMSE score.
The central finding of the manuscript is that Hcy is lowered post supplementation (Figure 5) and that MMSE is increased post supplementation (Figure 6). The real effect of B12 supplementation is somewhat hard to interpret given that all subjects were assessed after different lengths of supplementation.
- We have examined the different durations of supplementation as a confounding variable.
We had added the following description: “The confounder follow-up interval between starting vitamin B12 supplementation and MMSE reevaluation (P = 0.448), and the total follow-up period (P = 0.614) also had no effect on the change in the MMSE score.” (L222–224)
Perhaps most importantly, is an increase in of <3 in MMSE score a clinically meaningful change? This should be addressed. Similarly, the authors do not address the lack of specificity that is inherent in the MMSE.
-> We have added the following description: “The question of whether an improvement in the MMSE of <3 points is clinically significant remains. First, it was described that the MMSE has a specificity and sensitivity for predicting dementia of 82% and 87%, respectively [48]. In a previous randomized placebo-controlled trial of donepezil in patients with AD, the following conclusions were made. Improvements in the MMSE favoring donepezil were exhibited at weeks 6, 12, and 24 and at the end of the follow-up period. The improvements in the MMSE from the baseline value at weeks 6, 12, and 24 following the initiation of treatment were 1.4 (P = 0.02), 1.2 (P = 0.03), and 1.8 (P = 0.002) points, respectively. In that study, the placebo group exhibited an improvement in the MMSE of <0.3 points [49]. Donepezil is one of the gold-standard drugs used for treating AD. In this study, the improvement in the MMSE score exhibited following the treatment with vitamin B12 was 2.5 points (P < 0.0001; effect size, r = 0.73). Therefore, the change in the MMSE from vitamin B12 treatment can be significant.” (L302–313)
We also added the following description: “The MMSE score improved significantly during a short period from 20.5 ± 6.4 days to 22.9 ± 5.5 days following the vitamin B12 administration (P < 0.001; effect size, r = 0.73) (Figure 6). The power of the data following the vitamin B12 supplementation was 0.955” (L218–221).
Per your suggestion, the specificity and sensitivity of the MMSE in predicting dementia are 82% and 87%, respectively (Velayudhan L, et al, Int Psychiatrics 2014; 26; 1247–1262). We have added the following description in the text: “We estimated the cognitive function of patients by using only the MMSE, which has a specificity and sensitivity for predicting dementia of 82% and 87%, respectively [48].” (L341-343)
The application of the generalized mixed model is not clear. Were follow up MMSE measures taken at 0.5, 1, or 2 years? Is the GLMM used to predict this? This needs clarification in the methods section.
- We have added the following description: “A follow-up MMSE measurement was performed at 0.5, 1, or 2 years later (Supplementary Figure S4).” (L111-112)
The authors state in the discussion that this is a pre/post study without a control group. This should be stated in the abstract and/or introduction for full transparency.
- We have stated this in the abstract and introduction section using the following description: “In this multicenter, open-label, single-arm study.” (L33-34, 99, 267)
Minor concerns:
On page 2 (lines 53-63) the authors use “folic acid” in many places were “folate would be more appropriate.
-> We have revised the term “folic acid” to “folate” (L60, L62).
Also, on page 2 (lines 53-63), the authors state that B12 deficiency causes failure to convert dUMP to dTMP, which is true. However, it can also starve the cell of other nucleotide cofactors such as purines. Maybe more correct to just leave this as “tetrahydrofolate synthesis is impaired, which in turn impairs nucleic acid synthesis.”
->Consistent with your suggestions, we have changed the description to the following: “Vitamin B12 is not directly involved in nucleic acid synthesis; however, its deficiency results in the impairment of tetrahydrofolate synthesis, which in turn impairs nucleic acid synthesis [6-8].” (L61–63)
Page 2 (lines 75-76): probably more appropriate to say HHcy “is associated with” rather than “lead to” AD, VaD, etc.
- We have changed the description to the following: “HHcy is associated with AD [3, 6, 8-14], VaD [12, 15, 16], Parkinson's disease [17, 18], and stroke [12, 19, 20]” (L84–85).
Text says 1716 patients enrolled, figure 2 says 1715.
- We have amended Figure 2 to read “1716” (Figure 2).
Page 4 (lines 155-156): sentence starting “with folic acid” is not a complete sentence.
-> We have amended the manuscript as follows.
“Thirteen patients with concomitant folate deficiency [32] and 25 patients withdrew from the examination.” (L170-171)
Figure 7: In both panels the trend lines seem to be driven by a few outliers with exceedingly high Hcy levels. This should be addressed.
->We have added the following description: If the patients who exhibited a baseline Hcy level of >50 nmol/mL were removed, the change in the MMSE and the baseline Hcy level would exhibit no correlation (ρ=0.257, P=0.136). Furthermore, if the patients who exhibited a change in the Hcy level of >40 nmol/mL from the baseline level were removed, the change in the MMSE and the degree of decrease in the Hcy level would exhibit no correlation (ρ=0.120, P=0.551).” (L232–236)
Round 2
Reviewer 1 Report
The revision of this manuscript is fine, with one single exception that must be improved, the median and IQR being inadequately used.
E.g. l. 35: 21 - 133 days (median=56 days, IQR=36 days). It is not mentioned that the 21-133 is the full range. The median is fine, but IQR is defined by two values, not a single value like SD. The lower quartile value (p25) is a value between 21 and 56d (lower than 56), the upper value (p75) is a value between 56 and 133d (higher than 56).
The distance between p25 and median is different from the distance between median and p75, when data are not totally normal distributed. Hence, it cannot be a single value! I advise the authors to consult a statistician here!
Author Response
Reviewer1.
Thank you for your time and effort in reviewing our manuscript. We appreciate your helpful comments and believe that they have significantly improved the quality of our manuscript. We have revised the manuscript as outlined below.
Minor:
- The revision of this manuscript is fine, with one single exception that must be improved, the median and IQR being inadequately used.
E.g. l. 35: 21 - 133 days (median=56 days, IQR=36 days). It is not mentioned that the 21-133 is the full range. The median is fine, but IQR is defined by two values, not a single value like SD. The lower quartile value (p25) is a value between 21 and 56d (lower than 56), the upper value (p75) is a value between 56 and 133d (higher than 56).
The distance between p25 and median is different from the distance between median and p75, when data are not totally normal distributed. Hence, it cannot be a single value! I advise the authors to consult a statistician here!
-> Thank you very much for your comments. According to your suggestion, we have revised the manuscript as follows; (median=56 days, IQR=43-79 days) (L35). We have changed all of the IQR description as directed by the reviewer (L110, L173, L177, L178, L179-180, Table 1, L202-203, L266).
Reviewer 2 Report
I appreciate the many changes made by the authors and agree that this is an improved manuscript. I have one minor suggestion. The authors should add somewhere between lines 316-319 that when very high Hcy levels are removed the association between baseline Hcy and MMSE was no longer significant, as they have updated in the results section describing Figure 7A.
Author Response
Reviwer2.
Thank you for your time and effort in reviewing our manuscript. We appreciate your helpful comments and feel that they have significantly improved the quality of our manuscript. We revised the manuscript as outlined below.
Minor concern:
I appreciate the many changes made by the authors and agree that this is an improved manuscript. I have one minor suggestion. The authors should add somewhere between lines 316-319 that when very high Hcy levels are removed the association between baseline Hcy and MMSE was no longer significant, as they have updated in the results section describing Figure 7A.
-> Thank you very much for your positive comments. We have added the description at the discussion section as follows; However, if the patients who exhibited a baseline Hcy level of >50 nmol/mL were removed, the change in the MMSE and the baseline Hcy level would exhibit no correlation (L318-320).